# Machine learning meets maternal health: Uncovering spatial blind spots in antenatal care quality in Bangladesh

Sukanta Chakraborty ®*

Department of Statistics, University of Chittagong, Chittagong, Bangladesh

* chakraborty.sukanta@cu.ac.bd

## Abstract

### Background

High-quality antenatal care (ANC) is defined as four or more antenatal visits with at least one to a medically trained provider, measurement of weight and blood pressure, testing of blood and urine, and receipt of information on potential danger signs at least once during pregnancy. Though Bangladesh has almost universal ANC coverage, there is widespread inequality in the quality of these services. Traditional statistical models utilized in studies have tended to disregard complicated interconnections between socio-demographic, service-based, and regional factors that influence the quality of ANC. Using nationally representative data, this paper applies machine learning (ML) approaches to classify ANC quality, identify regional hotspots of low-quality care, and its factors.

### Methods

This study used data from the 2022 Bangladesh Demographic and Health Survey (BDHS), with a sample of 4587 women aged 15–49 who received ANC services. To predict binary ANC quality outcomes (high vs. low), three models were used: logistic regression, random forest (RF), and gradient boosting machine (GBM). Class imbalance was addressed using the ROSE (Random Over-Sampling Examples) technique, and model performance was evaluated using accuracy, sensitivity, specificity, and area under the ROC curve with 5-fold cross-validation. The most influential predictors were identified using feature importance analysis, and projected probabilities were aggregated at the cluster and division levels for spatial hotspot analysis. Geographic mapping was then utilized to demonstrate regional differences.

### Results

The GBM model outperformed the others, with the greatest prediction value (accuracy: 81.3%, sensitivity: 70.6%, specificity: 84.7%, AUC-ROC: 0.889). Number of

provided the original author and source are credited.

**Data availability statement:** The data underlying the results presented in the study are available from https://dhsprogram.com/data/available-datasets.cfm. Specifically, this study utilized the Bangladesh Demographic and Health Survey (BDHS) 2022 Individual Recode (IR) dataset, which includes information on ever-married women aged 15–49.

**Funding:** The author(s) received no specific funding for this work.

**Competing interests:** The authors have declared that no competing interests exist.

ANC visits, wealth index, place of residence, maternal education, and media access were all significant predictors. Spatial studies found hidden regions with high ANC visit coverage but low predicted ANC quality, highlighting considerable spatial differences in service quality. These hotspots are concentrated in Rangpur and Sylhet, which are far from Dhaka, the capital of Bangladesh, demonstrating spatial disparities in the usage of ANC services.

## Conclusions

The study shows that machine learning can classify ANC quality and reveal spatial disparities, aiding policymakers in targeting programs and allocating resources.

---

## Introduction

Antenatal care (ANC) is an important component of maternity and child health services that aims to reduce maternal morbidity and mortality by screening, preventing, and managing pregnancy-related problems. Comprehensive research by the United Nations (UN) Maternal Mortality Estimation Inter-Agency Group (MMEIG) found that in 2023, almost 700 women died every day worldwide as a result of pregnancy or delivery complications, with LDC nations accounting for around 43.9% of these deaths. According to World Health Organization (WHO) projections, pregnancy-related problems caused an astounding 2,60,000 fatalities in 2023; in Sub-Saharan Africa alone, these complications accounted for about 70% of all maternal deaths worldwide, with Central and Southern Asia accounting for nearly 17%. Approximately 7.2% of all maternal fatalities worldwide that year occurred in India alone [1]. However, Bangladesh has made significant progress, with a maternal mortality ratio of 115 deaths per 100,000 live births in 2023, yet the percentage of women with four or more antenatal care (ANC) visits declined from 46% in 2017–18–41% in 2022. These gaps in ANC coverage and quality continue to place women at higher risk of preventable complications such as hemorrhage, pre-eclampsia, and infections.

Reducing maternal deaths worldwide to less than 70 per 100,000 live births by 2030 is one of the most recent Sustainable Development Goals [2]. Appropriate ANC has been shown to reduce maternal mortality by up to 20% [3,4] and save lives [5]. Premature birth and low birth weight are two negative pregnancy outcomes that are prevented by high-quality ANC [6–8]. Women receiving comprehensive ANC are more likely to utilize skilled birth attendants—such as physicians, nurses, and midwives—during delivery and postpartum care [9]. Additionally, appropriate ANC can reduce maternal mortality and improve overall pregnancy outcomes.

The 2016 WHO ANC model recommends several interventions to ensure high-quality care, including nutrition counseling, maternal and fetal assessments, preventive measures, management of physiological symptoms, and strengthening health system support [2]. In Bangladesh, surveys have assessed whether women received these essential services during ANC, such as blood pressure and weight

measurements, blood and urine testing, ultrasounds, counseling on pregnancy complications, and postpartum family planning.

The idea of quality ANC has been defined using these criteria in conjunction with the requirement that a minimum of four visits be completed during ANC [10,11–13]. However, only 21% receive quality ANC in Bangladesh, as documented by the BDHS 2021–22. 41% had the recommended four or more ANC visits during their pregnancy. Women in urban areas (57%) were more likely to have four or more visits than women in rural areas (35%) [14]. Among mothers with at least one ANC visit, more than 90% had their blood pressure and weight measured; 80%, 81%, and 94% had a blood test, urine test, and ultrasound, respectively; 50% received information on danger signs during pregnancy; and 27% received information on postpartum family planning [14].

While antenatal care (ANC) coverage has significantly increased in low- and middle-income countries (LMICs) like Bangladesh, notable disparities persist in the quality of services provided to expectant mothers. For instance, a study utilizing data from the 2017–18 Bangladesh Demographic and Health Survey (BDHS) found that only 21% of women received quality ANC services, defined as attending at least four ANC visits with essential services such as blood pressure and weight measurements, blood and urine testing, ultrasounds, and counseling on pregnancy complications [15]. Additionally, rural women were 12% less likely than urban women to receive the recommended number of ANC visits, highlighting a significant urban-rural disparity [16]. So, there are large gaps in the quality of ANC received according to women's background characteristics. For example, women in urban areas are twice as likely to receive quality ANC as women in rural areas (33% versus 17%). Only 8% of women in the lowest wealth quintile receive quality ANC, compared with 39% of women in the highest quintile [14].

Although ANC coverage in Bangladesh is almost widespread, questions still surround the sufficiency and caliber of these services, particularly for underprivileged groups. In Bangladesh, there aren't many studies on ANC, and most of them were carried out in rural areas [17–22]. Data on ANC is gathered once every three years through the nationally representative BDHS survey. Few scholars have looked at the causes and contents of ANC in Bangladesh using the demographic and health survey data that is currently available [23–26]. Rahman et al. (2017) used the BDHS 2011 and 2014 to investigate the factors that influence the use of four or more ANCs. They discovered that wealth status, place of residence, and educational attainment were the main determinants [23]. To the best of the authors' knowledge, the improvement of the quality of ANC from 2014 to 2017–18 is studied by Akter et al. (2023) [27].

Prior research has mostly concentrated on ANC coverage measurement, paying little attention to evaluating ANC quality or finding social and geographic "blind spots" when high service consumption is accompanied by poor ANC quality. Closing this gap is essential to aligning with the Sustainable Development Goals (SDGs) and guaranteeing equitable maternal health outcomes.

To identify women at risk of receiving inadequate ANC, we employ machine learning (ML) techniques. While traditional regression models can also perform this task, ML offers several advantages, such as the ability to handle high-dimensional data, capture non-linear relationships, and model complex interactions between socioeconomic, demographic, and geographic factors [28]. Recent studies in maternal and child health have demonstrated that ML approaches often outperform conventional models in predicting health service utilization and maternal outcomes [29,30], making them well-suited for uncovering hidden patterns in ANC quality. Accordingly, we apply ML techniques to map these "hidden hotspots"—regions where women are going to ANC visits but aren't getting the care they need.

This study, which combines state-of-the-art analytics with geospatial tools, not only pinpoints the locations of issues but also explains why they continue to exist, such as poverty, education, access issues, or other social injustices. Our objective is to give health leaders and policymakers practical, understandable information to enhance maternal health care where they are most needed.

To ensure that no mother is left behind, regardless of her location, this research aims to shift the focus of the discussion from the number of women who receive ANC to the quality of care they receive.

 

## Methods

### Data source

This study examined data from the 2022 BDHS, a nationally representative dataset. Presenting the most recent estimates of demographic and health variables, particularly fertility, childhood mortality, maternal and child health, nutrition, and neonatal care, and to grow awareness, approval, and use of family planning methods, is the aim of this national survey. The National Institute of Population Research and Training (NIPROT) and the Ministry of Health and Family Welfare, Government of the People's Republic of Bangladesh, oversee the BDHS, a cross-sectional survey carried out by Mitra and Associates. The final summary report of the 2022 BDHS includes a thorough explanation of the survey design, procedures, sample size, and questionnaires.

### Study population and survey design

The survey is based on a stratified two-stage sample of households. Initially, 675 enumeration areas (EAs) were selected, 237 of which were in urban areas and 438 of which were in rural areas, using a probability based on EA size. To provide statistically reliable estimates of key health and demographic characteristics for each of the eight divisions, for the country overall, and for urban and rural areas individually, a systematic sample of 45 households on average per EA was selected for the second sampling step. On the basis of this idea, 30,375 residential properties (19,710 from rural areas and 10,665 from urban areas) were selected. Women aged 15–49 who had a live birth or stillbirth in the 2 years preceding the survey have been analyzed. Interviews were completed with 30,078 women, yielding a response rate of 99.1%. After data preprocessing, we get a sample of 4587 women.

### Dependent variable

Quality ANC is the dependent variable in our study. To have quality antenatal care (ANC), a woman should experience four or more antenatal visits with at least one to a medically trained provider, measurement of weight and blood pressure, testing of blood and urine, and receipt of information on potential danger signs at least once during pregnancy. So, in our study, we first make a binary variable (0 = low quality, 1 = high quality). Women who received all the above antenatal care services were classified as having high-quality ANC, and women who did not receive all the above antenatal care services were classified as having low-quality ANC.

### Independent variables

The selection of explanatory variables for our study was based on prior research on the quality of maternal and neonatal care services as well as the structure of BDHS reports. Respondent's current age, education level of the participants and their husbands (categorized as no education, primary, secondary, higher), residence (urban, rural), division (Dhaka, Chattogram, Rajshahi, Sylhet, Barisal, Khulna, Rangpur, Mymensingh), wealth quintile (categorized as poorest, poorer, middle, richer, richest), currently working status of the participants(yes/no), religion, cluster altitude, access to mass media (yes/no), a autonomy variable about person who usually decides on respondent's health care, place of taking ANC service (home, public hospital, private hospital) and no of ANC visits are considered as explanatory variables in the study.

### Statistical methods and machine learning models

Sophisticated machine learning (ML) approaches such as Random Forest and Gradient Boosting Machine (GBM) have been applied, alongside conventional statistical modeling approaches like Logistic Regression (LR), to predict the risk of low-quality antenatal care (ANC) and identify key factors associated with these risks.

**Logistic regression.** For classification problems, the binomial Generalized Linear Model (GLM), often known as logistic regression, is a popular statistical model. To estimate the parameters of interest, maximum likelihood estimation

is used. In addition to its use in population health research as an inferential tool, logistic regression is a useful technique for binary categorization. Since it assumes a logit relationship between the response variable and predictors and does not call for hyperparameter modification, this model is especially well-suited for evaluating binary data [31].

**Random forest.** An ensemble learning technique that specializes in decision tree ensembles is Random Forest (RF), sometimes referred to as a decision tree forest. By building numerous deep trees from bootstrap data, it improves model stability and lowers variance as a bagging strategy. The foundation of RF is ensemble learning, which combines several classifiers to enhance prediction accuracy and address challenging issues. By randomly selecting characteristics for each tree, RF lowers the correlation between trees in contrast to typical decision trees that sample from the full dataset. A distinct subset of predictor variables is used to train each tree, and the majority vote of all trees is used to make predictions. This technique reduces overfitting and increases accuracy, particularly as the number of trees increases [32].

We set two hyperparameters: the number of trees (ntree) and the number of variables we tried at each split (mtry). We evaluated ntree values from 100 to 500 and found that ntree = 100 stabilized the model while limiting computation time. We adjusted mtry to different values around the default ($\sqrt{p}$) and chose the one that gave us the lowest out-of-bag (OOB) error rate, where p is the number of predictors. To ensure robustness and prevent overfitting, we applied 5-fold cross-validation during hyperparameter selection.

**Gradient boosting machine (GBM).** A potent ensemble learning system, the Gradient Boosting Machine (GBM) combines several weak learners to produce a strong learner. In order to arrive at the final prediction, it iteratively fits new models to the residuals of earlier models. Compared to other machine learning algorithms, GBM has a number of benefits, including the ability to handle mixed data types, automatic feature selection, and resilience to missing data and outliers [33].

The model was trained using 100 boosting iterations (n.trees = 100), allowing it to improve over time. Setting the interaction depth to 3 allowed predictors in each tree to interact in up to three different ways. The Bernoulli loss function was used, which is suitable for binary classification. Hyperparameters, including n.trees (50–200), interaction. depth (1–5), and learning rate (shrinkage, 0.01–0.1), were tuned using 5-fold cross-validation to identify the combination that optimized ROC-AUC on the training set.

## Data analysis

**Data preprocessing.** All categorical variables were transformed into factor variables. While missing values in predictor variables were kept, observations with missing values in the dependent variable (ANC quality) were eliminated; exploratory checks showed that missing data in predictors had no noticeable effect on model training or performance, hence no imputation was used.

The outcome variable, ANC quality, was binary (High vs. Low), and the training set showed a moderate class imbalance (~24% High vs. 76% Low). To solve this, the ROSE (Random Over-Sampling Examples) technique was used on the training set to generate synthetic samples from the minority class while gently influencing the majority class. Logistic Regression, Random Forest, and Gradient Boosting Machine models were trained on this balanced dataset, and their performance was tested on the original, unbalanced test set using accuracy, sensitivity, specificity, and ROC-AUC to ensure that the results reflected the real-world class distribution. Descriptive statistics were used to assess the sociodemographic characteristics of the targeted population. Logistic regression, Random Forest, and Gradient Boosting Machine (GBM) models were fitted to the study data.

**Analysis of feature importance and model evaluation.** Two techniques were employed to assess the significance of each feature: Random Forest and GBM Models were used to determine the most significant factors influencing the classification of ANC quality, and variable importance scores were retrieved. It should be mentioned that rather than representing a causal effect, these significance scores represent a predictive contribution. Logistic Regression was used to measure the degree and direction of relationships between the predictors and ANC quality; odds ratios (ORs) and 95%

CIs were computed. To assess the performance of the models, the Area Under the Receiver Operating Characteristic Curve (AUC-ROC) was used.

**Hotspot and spatial analysis.** Hotspot analysis processes included aggregating the projected probability of low ANC quality at the cluster and division levels in order to look for spatial differences in ANC quality.

The term "hidden hotspots" refers to clusters that have a high number of ANC visits (ANC visits ≥ 4) but a reduced expected likelihood of receiving high-quality ANC, specifically the lowest 40% quartile of predicted probability. The 40th percentile threshold was chosen to strike a balance between sensitivity and specificity in detecting at-risk clusters; more restrictive thresholds may ignore susceptible areas, whereas more permissive thresholds may contain clusters at moderate risk. Estimates for clusters with lower sample sizes may be more variable, potentially compromising the stability of the estimated probability. Cluster-level aggregation was used to provide meaningful spatial interpretation while reducing the influence of individual-level random variation. This strategy essentially highlights the regions where women can get ANC services, yet have obstacles to receiving high-quality care. This strategy essentially highlights the regions where women can get ANC services, yet have obstacles to receiving high-quality care. The data were analyzed using the R programming language on R Studio and SPSS 16.0.

**Ethical consideration.** This study used de-identified, publicly available data from the 2022 BDHS. Therefore, no extra institutional ethical approval was needed.

## Results and discussion

### Distribution of ANC quality concerning demographic characteristics

The relationship between antenatal care (ANC) quality and various socio-demographic, economic, and healthcare-related factors of the study participants included in this study is presented in Table 1. High-quality ANC increased with maternal age, peaking at 27.6% among women aged 35–39, and was absent among women aged 45–49. Regionally, Mymensingh (28.9%) and Dhaka (27.2%) had the highest coverage, whereas Rangpur (18.0%) and Sylhet (19.7%) had the lowest. Urban women were more likely to receive high-quality ANC (32.9%) than rural women (19.1%).

Educational attainment and wealth were positively associated with ANC quality. Women with higher education (41.5%) and those from the richest households (40.6%) were most likely to receive high-quality ANC. Access to mass media also favored higher-quality care (28.6% vs. 16.8% without access).

Use of private facilities modestly increased high-quality ANC (26.0% vs. 20.3%), while home or public facility visits had little effect. Shared decision-making with husbands was linked to better care (25.7% vs. 18.9% when husbands decided alone).

The number of ANC visits was a strong determinant: women with fewer than four visits universally received low-quality care, whereas 53.4% of women with four or more visits received high-quality ANC. These findings highlight persistent socioeconomic, educational, and service-related disparities in ANC quality across Bangladesh.

### Quality ANC

The quality of ANC services that the mothers received, as analyzed based on the seven core ANC components, is reported in Fig 1. Compared to other components, pregnancy-related counseling during ANCs was shown to be lower. Of the mothers, 44.5% have 4 or more ANC visits & only half (50%) had received counseling regarding pregnancy-related warning indicators, while almost more than two-thirds (82.8%) and 79.9%, respectively, underwent urine and blood testing. Nonetheless, during ANCs, the majority of subjects had their blood pressure, weight measured and went through ultrasonography (91.3%, 93% and 93.7%, respectively). Just 23.8% of quality ANC services were provided overall.

### Association between demographic characteristics and outcome variables

Table 2 shows the adjusted multivariable logistic regression model for the association between the quality of ANC and predictor variables. Compared to women in Barisal, those in the Mymensingh division were statistically significantly

**Table 1. Distribution of outcomes by ANC quality and background characteristics of women aged 15-49 who had a live birth or stillbirth in the 2 years preceding the BDHS 2022 survey.**

| | | Binary ANC Quality | |
|---|---|---|---|
| | | Low quality | High quality |
| Age in 5-year groups | 15-19 | 525(83.1%) | 107(16.9%) |
| | 20-24 | 1165(77.6%) | 337(22.4%) |
| | 25-29 | 941(73.7%) | 336(26.3%) |
| | 30-34 | 586(73.8%) | 208(26.2%) |
| | 35-39 | 231(72.4%) | 88(27.6%) |
| | 40-44 | 43(75.4%) | 14(24.6%) |
| | 45-49 | 6(100.0%) | 0(0.0%) |
| Division | Barishal | 386(76.7%) | 117(23.3%) |
| | Chattogram | 610(77.3%) | 179(22.7%) |
| | Dhaka | 503(72.8%) | 188(27.2%) |
| | Khulna | 414(76.2%) | 129(23.8%) |
| | Mymensingh | 392(71.1%) | 159(28.9%) |
| | Rajshahi | 358(74.1%) | 125(25.9%) |
| | Rangpur | 438(82.0%) | 96(18.0%) |
| | Sylhet | 396(80.3%) | 97(19.7%) |
| Place of residence | Urban | 1043(67.1%) | 511(32.9%) |
| | Rural | 2454(80.9%) | 579(19.1%) |
| Respondent's educational level | No education | 165(85.9%) | 27(14.1%) |
| | Primary | 843(85.7%) | 141(14.3%) |
| | Secondary | 1941(78.4%) | 534(21.6%) |
| | Higher | 548(58.5%) | 388(41.5%) |
| Religion | Islam | 3196(76.4%) | 988(23.6%) |
| | Hinduism | 276(74.2%) | 96(25.8%) |
| | Buddhist | 18(90.0%) | 2(10.0%) |
| | Christianity | 7(63.6%) | 4(36.4%) |
| Access to mass media | No | 1560(83.2%) | 314(16.8%) |
| | Yes | 1937(71.4%) | 776(28.6%) |
| Wealth index | Poorest | 750(89.0%) | 93(11.0%) |
| | Poorer | 751(83.2%) | 152(16.8%) |
| | Middle | 731(78.9%) | 195(21.1%) |
| | Richer | 692(72.8%) | 258(27.2%) |
| | Richest | 573(59.4%) | 392(40.6%) |
| Received ANC from the respondent's home | No | 3274(76.2%) | 1020(23.8%) |
| | Yes | 223(76.1%) | 70(23.9%) |
| Received ANC from any public health facility | No | 2306(75.6%) | 743(24.4%) |
| | Yes | 1191(77.4%) | 347(22.6%) |
| Received ANC from any private health facility | No | 1437(79.7%) | 367(20.3%) |
| | Yes | 2060(74.0%) | 723(26.0%) |
| Husband/partner's education level | No education | 515(83.6%) | 101(16.4%) |
| | Primary | 1131(85.5%) | 192(14.5%) |
| | Secondary | 1246(76.3%) | 387(23.7%) |
| | Higher | 605(59.6%) | 410(40.4%) |
| Respondent currently working | No | 2749(75.9%) | 874(24.1%) |
| | Yes | 748(77.6%) | 216(22.4%) |

*(Continued)*

| | | Binary ANC Quality | |
|---|---|---|---|
| | | **Low quality** | **High quality** |
| The person who usually decides on: the respondent's health care | Respondent alone | 303(75.6%) | 98(24.4%) |
| | Respondent and husband | 2164(74.3%) | 750(25.7%) |
| | Husband/partner alone | 897(81.1%) | 209(18.9%) |
| | Someone else | 122(79.2%) | 32(20.8%) |
| | Other | 11(91.7%) | 1(8.3%) |
| No of ANC visits | less than 4 | 2544(100%) | 0(0%) |
| | 4 or more | 953(46.6%) | 1090(53.4%) |

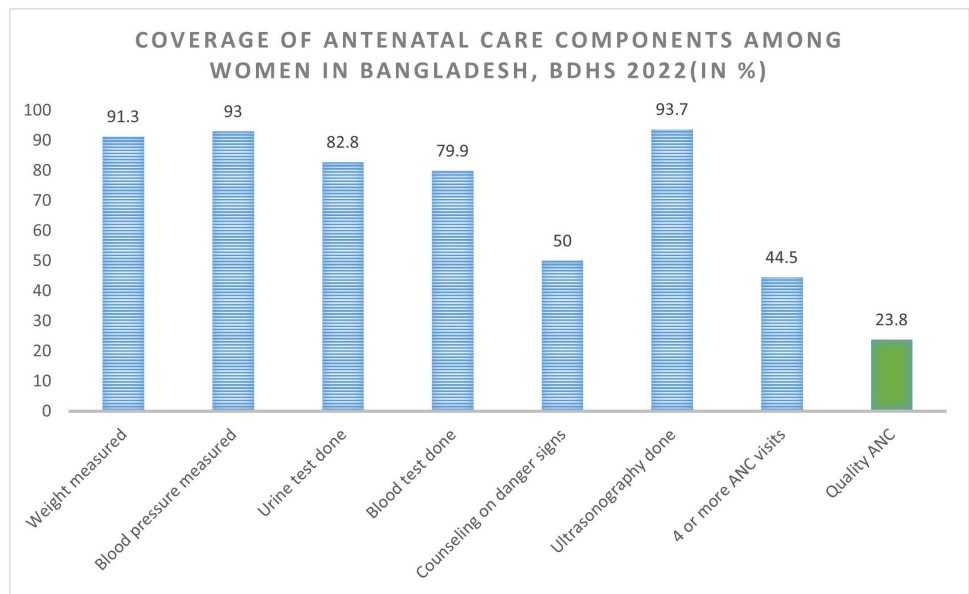

**Fig 1. The six components of quality ANC as received by women aged 15-49 years who had a live birth or stillbirth in the 2 years preceding the BDHS 2022 survey, (%).**

more likely to receive high-quality ANC (OR: 1.83; p = 0.038). Compared to urban women, rural women were statistically significantly less likely to obtain high-quality ANC (OR: 0.747; p = 0.019). Additionally, there was a statistically significant negative correlation between cluster altitude and ANC quality (OR: 0.977; p = 0.002). In terms of religion, Hindu women exhibited a lower likelihood than Muslim women to receive high-quality ANC (OR: 0.656; p = 0.039), and the odds were even lower for Buddhist women (OR: 0.152; p = 0.028).

ANC quality was shown to be strongly predicted by wealth class. Women in the poorer (OR: 1.471; p = 0.047), middle (OR: 1.633; p = 0.013), and richer (OR: 1.906; p = 0.001) groups were markedly more likely to receive high-quality ANC than those in the poorest group. Additionally, the power of decision-making was important: women who collaborated with their husbands on healthcare decisions had higher odds of receiving high-quality ANC than those who made decisions alone (OR: 1.529; p = 0.020). The most significant predictor was the quantity of ANC visits; women who had more ANC visits were more than twice as likely to have high-quality ANC (OR: 2.025; p < 0.001).

**Table 2. Association between outcomes and exposures, OR (95% CI).**

| Background characteristics | OR | 95% Confidence Interval | | p – value |
|---|---|---|---|---|
| | | Lower | Upper | |
| Age | 1.01 | 0.991 | 1.029 | 0.286 |
| Division | | | | |
| Barisal | Ref. | Ref. | Ref. | Ref. |
| Chattogram | 1.211 | 0.732 | 2.054 | 0.466 |
| Dhaka | 0.853 | 0.51 | 1.458 | 0.551 |
| Khulna | 0.807 | 0.461 | 1.435 | 0.458 |
| Mymensingh | 1.83 | 1.042 | 3.275 | 0.038 |
| Rajshahi | 1.384 | 0.778 | 2.506 | 0.275 |
| Rangpur | 1.49 | 0.724 | 3.095 | 0.281 |
| Sylhet | 1.017 | 0.527 | 1.967 | 0.960 |
| Residence | | | | |
| Urban | Ref. | Ref. | Ref. | Ref. |
| Rural | 0.747 | 0.587 | 0.953 | 0.019 |
| Cluster altitude | 0.977 | 0.963 | 0.991 | 0.002 |
| Respondent's Education | | | | |
| No Education | Ref. | Ref. | Ref. | Ref. |
| Primary | 0.848 | 0.483 | 1.527 | 0.574 |
| Secondary | 0.811 | 0.467 | 1.445 | 0.466 |
| Higher | 1.026 | 0.558 | 1.926 | 0.935 |
| Religion | | | | |
| Islam | Ref. | Ref. | Ref. | Ref. |
| Hinduism | 0.656 | 0.436 | 0.972 | 0.039 |
| Buddhist | 0.152 | 0.024 | 0.739 | 0.028 |
| Christianity | 3.185 | 0.493 | 16.777 | 0.186 |
| Access to mass media | | | | |
| No | Ref. | Ref. | Ref. | Ref. |
| Yes | 1.229 | 0.981 | 1.543 | 0.074 |
| Wealth Index | | | | |
| Poorest | Ref. | Ref. | Ref. | Ref. |
| Poorer | 1.471 | 1.008 | 2.16 | 0.047 |
| Middle | 1.633 | 1.113 | 2.413 | 0.013 |
| Richer | 1.906 | 1.292 | 2.834 | 0.001 |
| Richest | 1.39 | 0.897 | 2.163 | 0.143 |
| ANC service from home | | | | |
| No | Ref. | Ref. | Ref. | Ref. |
| Yes | 0.8 | 0.528 | 1.194 | 0.282 |
| ANC service from the public hospital | | | | |
| No | Ref. | Ref. | Ref. | Ref. |
| Yes | 1.185 | 0.926 | 1.517 | 0.177 |
| ANC service from a Private hospital | | | | |
| No | Ref. | Ref. | Ref. | Ref. |
| Yes | 1.26 | 0.985 | 1.613 | 0.066 |
| Husband's education | | | | |
| No Education | | | | |
| Primary | 0.759 | 0.533 | 1.084 | 0.127 |

*(Continued)*

**Table 2.** (Continued)

| Background characteristics | OR | 95% Confidence Interval | | p – value |
|---|---|---|---|---|
| | | Lower | Upper | |
| Secondary | 0.838 | 0.589 | 1.199 | 0.330 |
| Higher | 0.904 | 0.596 | 1.374 | 0.635 |
| Respondent's working status | | | | |
| No | Ref. | Ref. | Ref. | Ref. |
| Yes | 0.925 | 0.716 | 1.19 | 0.545 |
| Decision about health care | | | | |
| Respondent alone | Ref. | Ref. | Ref. | Ref. |
| Respondent and husband/partner | 1.529 | 1.075 | 2.202 | 0.020 |
| Husband/partner alone | 1.252 | 0.836 | 1.89 | 0.280 |
| Someone else | 1.443 | 0.753 | 2.704 | 0.259 |
| Other | 0.368 | 0.008 | 3.998 | 0.500 |
| No of ANC Visits | 2.025 | 1.903 | 2.159 | 0.000 |

Other variables, such as the education of the respondent and spouse, media access, work, and the kind of healthcare institution, did not exhibit any noteworthy correlations following adjustment. More information can be found in supplementary file(S1 table).

## Analysis of feature importance by random forest and gradient boosting machine models

Fig 2 depicts the comparative feature importance from Random Forest and Gradient Boosting Machine (GBM) models for high-quality antenatal care (ANC) prediction.

The primary predictor of ANC quality in both models was the quantity of ANC visits (No of ANC Visits). However, in the Random Forest model, No of ANC Visits' relevance value was more than that of every other variable taken into account, highlighting the feature's overwhelming contribution to prediction accuracy; Despite having a smaller degree of impact than Random Forest, the GBM model likewise found No of ANC Visits to be the most influential attribute. Cluster altitude, Division, Wealth index, and Age are additional variables of moderate importance; this suggests that demographic, socio-economic, and geographic factors have a moderate impact on the ANC quality. Residence Area, Husband's Education, and Access to Mass Media rank lower in the Random Forest model but still make a modest contribution. Interestingly, both models gave little weight to a number of characteristics, including religion, autonomy, and the ANC service points that were taken into consideration (home, public, and private). This suggests that these variables would have very little predictive value for ANC quality in this configuration. It is important to emphasize that the importance scores derived from the machine learning models are interpreted as predictive contributions rather than causal relationships; the variables identified are considered markers of prediction rather than determinants of ANC quality. These findings suggest that, in addition to increasing ANC visits, attention should be given to structural, geographic, and socioeconomic factors to improve antenatal care quality.

## Model comparison between traditional and machine learning models

Table 3 and Fig 3 show some significant variations in prediction performance by the model comparison matrices. With an overall accuracy of 81.3%, the Gradient Boosting Machine (GBM) model outperformed the other two models. Logistic Regression received the lowest score at 80.0%, followed by Random Forest at 81%. Out of the three models, GBM was the most effective in accurately identifying high-quality ANC cases, as evidenced by its highest sensitivity of 70.6%. The best specificity was achieved using logistic regression, which identified low-quality ANC cases with a 91.5% accuracy rate. In the AUC-ROC analysis, the three models performed reasonably similarly, with GBM showing a tiny advantage (0.8888).

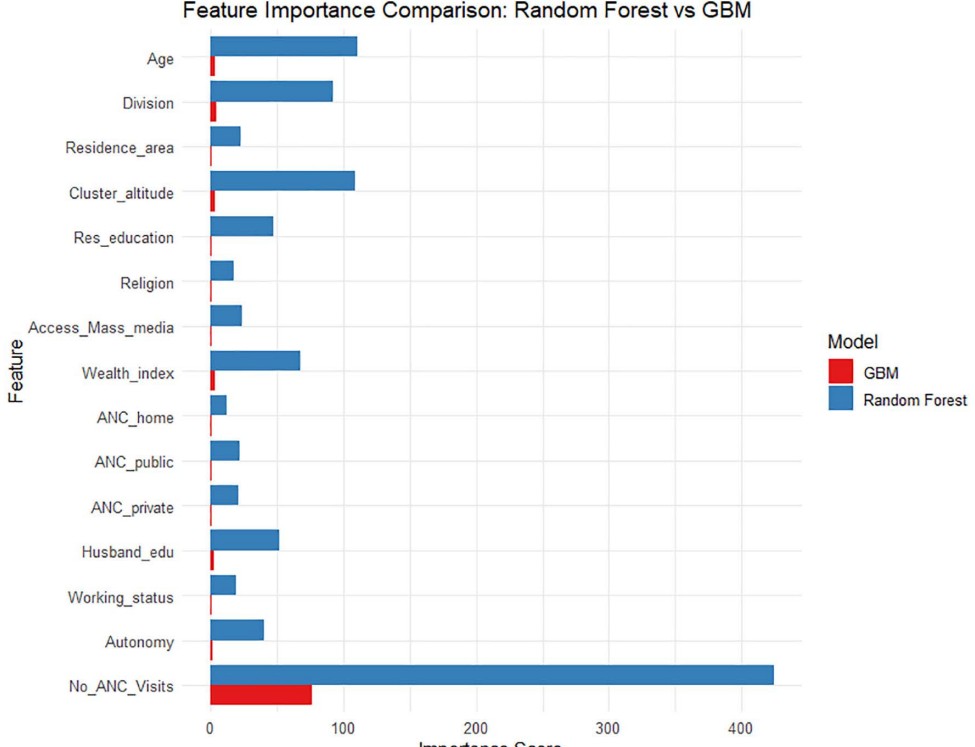

**Fig 2. Feature importance comparison between the machine learning algorithms used in this study.**

**Table 3. Model Evaluation Matrices.**

| Model Name / Evaluation Matrix | Logistics | Random Forest | Gradient Boosting Machine |
|---|---|---|---|
| Accuracy | 0.8001 | 0.8096 | 0.8132 |
| Sensitivity | 0.4312 | 0.6177 | 0.7064 |
| Specificity | 0.9152 | 0.8694 | 0.8465 |
| ROC-AUC | 0.8761 | 0.8882 | 0.8888 |

After taking into consideration all the matrices, GBM is the best model for its overall predictive power, while Random Forest is competitive, having a better sensitivity as compared to Logistic Regression.

## Hotspot analysis

Table 4 and Fig 4 show some hotspots of having low-quality ANC services with a higher number of ANC visits.

Using the GBM model, the likelihood that each woman will receive high-quality prenatal care (ANC) was calculated. To determine whether clusters were deemed hotspots—Clusters with a high average number of ANC visits (≥4 visits) but low predicted quality of ANC (below the 40th percentile)—the estimated probabilities were then aggregated at the cluster level.

Six (06) such clusters were found, with a low estimated mean probability of 0.103 to 0.170 for the provision of high-quality ANC, and an average maximum of 5 ANC visits and 4.0 for the minimum (with WHO minimum standards). Two to eight women were questioned for each cluster.

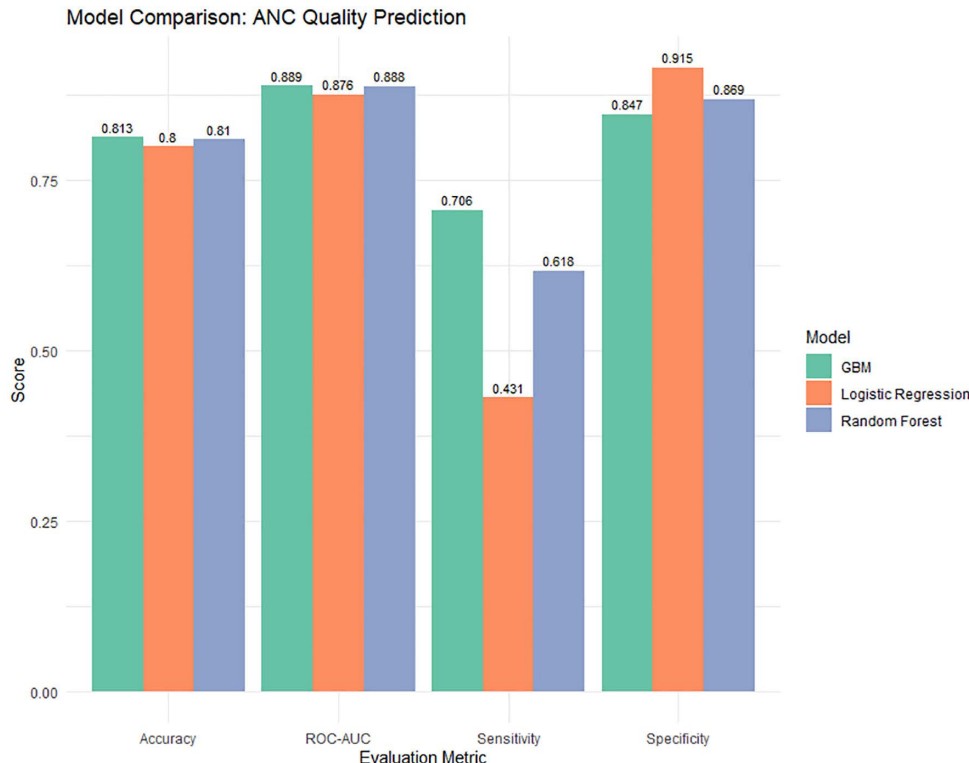

**Fig 3. Graphical comparison of the fitted model by evaluation matrices.**

**Table 4. Hotspot analysis of Low ANC quality.**

| Sl. No. | Cluster number | Mean predicted probability of high-quality ANC | Mean no of ANC visits | No of women interviewed in the cluster | No of hotspots in each cluster |
|---------|----------------|-----------------------------------------------|-----------------------|----------------------------------------|-------------------------------|
| 1 | 28 | 0.151 | 4 | 5 | 1 |
| 2 | 71 | 0.154 | 5 | 2 | 1 |
| 3 | 195 | 0.133 | 4.2 | 5 | 1 |
| 4 | 311 | 0.103 | 4.2 | 5 | 1 |
| 5 | 326 | 0.167 | 4.33 | 6 | 1 |
| 6 | 600 | 0.170 | 4.5 | 8 | 1 |

This analysis suggests that, despite high ANC utilization in many regions, care quality may still be lacking. These clusters significantly hamper the achievement of maternal health outcomes through quality improvement measures.

## Spatial analysis of antenatal care quality in Bangladesh

Fig 5 depicts a regional map of the projected probability of high-quality prenatal care (ANC) across Bangladesh's eight divisions, as derived from a predictive modeling study. The anticipated probabilities range between 0.183 and 0.277, with considerable geographical differences. Mymensingh (0.277) and Dhaka (0.271) had the greatest anticipated probabilities, indicating that these divisions have significantly better coverage of excellent ANC services. The converse is true in Rangpur (0.183) and Sylhet (0.202), which were projected with the lowest probabilities, indicating locations where quality

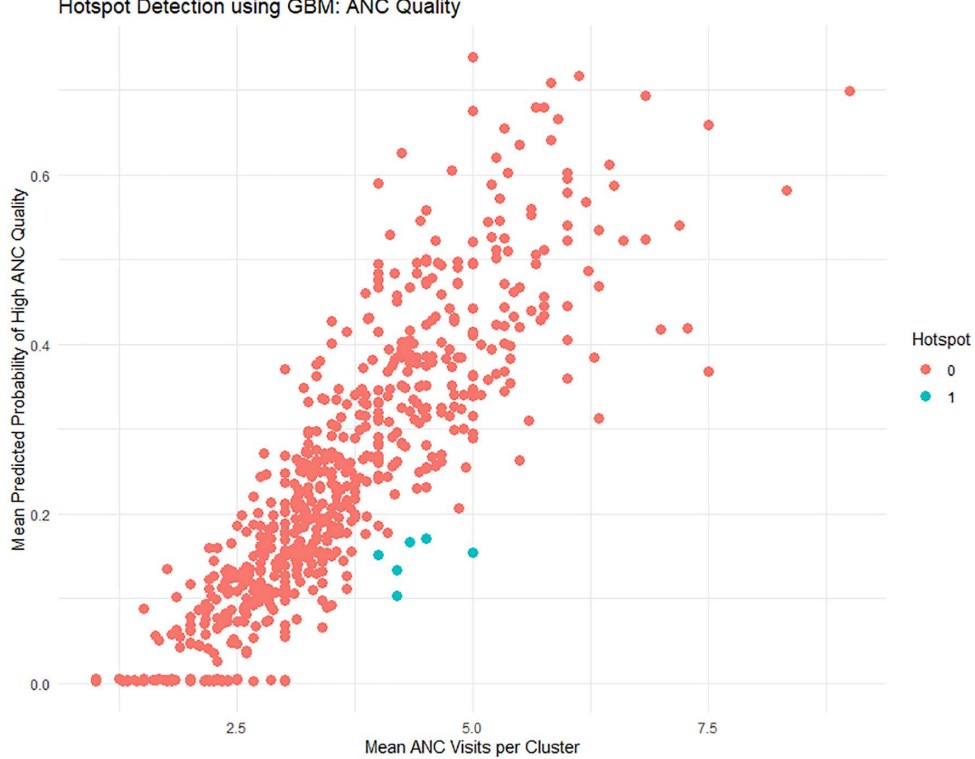

**Fig 4. Hotspot detection of low ANC quality.**

ANC coverage may be absent. The projected probabilities for Rajshahi (0.265), Chattogram (0.246), Khulna (0.244), and Barishal (0.222) were all intermediate.

The spatial clustering of greater probabilities around the central divisions (Dhaka and Mymensingh) and lower probabilities in the northern to northeastern region (Rangpur and Sylhet) sheds additional light on the geographic variance in expected ANC quality. Such regional heterogeneity may reflect variations in healthcare infrastructure, socioeconomic status, and maternal health service consumption. It is therefore appropriate to target poor-performing areas for enhanced policy focus on improving ANC services and decreasing regional disparity.

## Strengths and limitations

This is the first study to uncover the spatial blind spot through machine learning algorithms. The fact that this study used demographic survey data ensures that the women included are nationally representative, improving the findings' generalizability across a range of demographics. As a result, this extensive dataset provides a trustworthy standard for further study. Furthermore, this study offers important insights to guide policy and actions targeted at enhancing the quality of antenatal care services by addressing important maternal health variables and finding hotspots of low ANC quality.

The identification of hotspots of low ANC quality enables more precise targeting of interventions. By highlighting specific geographic areas where women have access to ANC services but are at elevated risk of receiving suboptimal care, resources can be allocated efficiently to improve service delivery, training, and supervision. Additionally, hotspot analysis facilitates ongoing monitoring, allowing policymakers to track the impact of interventions over time and adjust strategies to address persistent gaps. This approach ensures that both structural and service-level challenges are addressed in a data-driven manner, ultimately supporting efforts to enhance maternal health outcomes.

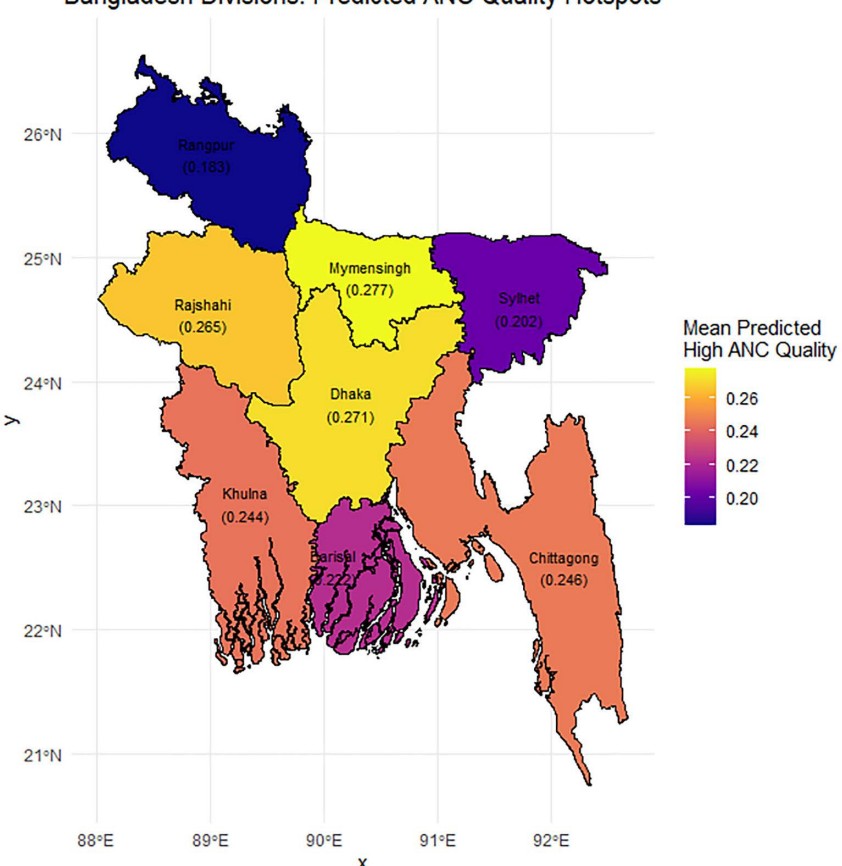

Bangladesh Divisions: Predicted ANC Quality Hotspots

**Fig 5. The predicted hotspots of high-quality antenatal care (ANC) across divisions of Bangladesh.** Source: Adapted from https://data.humdata.org/dataset/geoboundaries-admin-boundaries-for-bangladesh, licensed under CC BY 4.0.

This study is subject to a number of limitations. There might have been recall bias. Additionally, some women may have misidentified the health facility or provider where they obtained ANC services. Moreover, the cross-sectional design of the BDHS limits the ability to infer temporal or causal relationships. Unmeasured confounding variables, such as provider competence or facility-level resources, may also have influenced the observed associations, potentially affecting the interpretation of the results.

### Future research opportunities

Causal ML approaches like Double Machine Learning (DML) and Targeted Maximum Likelihood Estimation (TMLE) can be used to evaluate the causal effects of predictors on ANC quality. Moreover, one can integrate ML models with mobile health tools to give early warning systems for frontline workers.

### Conclusion

To increase the quality of ANC services, it is important to start with underserved locations, such as Rangpur and Sylhet, which face the greatest shortages, and to prioritize the needs of vulnerable populations. Strengthening facility-based services through improved staff training and supervision, expanding outreach and health education initiatives, improving infrastructure and accessibility, and developing monitoring tools to track quality improvements can help address persistent

gaps in maternal healthcare. Without such targeted efforts, improving ANC quality in Bangladesh will remain a significant challenge.

## Supporting information

**S1 Table:  Association between outcomes and predictors, p-value.**
(DOCX)

## Acknowledgments

The authors would like to express their gratitude to all of the men and women who took part in the 2022 Bangladesh Demographic and Health Survey.

## Author contributions

**Conceptualization:** Sukanta Chakraborty.

**Data curation:** Sukanta Chakraborty.

**Formal analysis:** Sukanta Chakraborty.

**Methodology:** Sukanta Chakraborty.

**Writing – original draft:** Sukanta Chakraborty.

**Writing – review & editing:** Sukanta Chakraborty.

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
