## [Decision Letter · Decision Letter 0]

22 Sep 2025

Dear Dr. Chakraborty,

The variable selection process needs better justificationThe study lacks proper cross-validation. Please include a cross-validation process to ensure that there is no overfitting concerns.Only 23.8% receiving ANC, the data is quite imbalanced. Consider adding some information on the sampling processAdditional information on how machine learning makes predictions better.

We look forward to receiving your revised manuscript.

Kind regards,

Russell Kabir, PhD

Academic Editor

PLOS ONE

Journal Requirements:

2. We note that Figure 5 in your submission contain map images which may be copyrighted. All PLOS content is published under the Creative Commons Attribution License (CC BY 4.0), which means that the manuscript, images, and Supporting Information files will be freely available online, and any third party is permitted to access, download, copy, distribute, and use these materials in any way, even commercially, with proper attribution. For these reasons, we cannot publish previously copyrighted maps or satellite images created using proprietary data, such as Google software (Google Maps, Street View, and Earth). For more information, see our copyright guidelines: http://journals.plos.org/plosone/s/licenses-and-copyright.

a. You may seek permission from the original copyright holder of Figure 5 to publish the content specifically under the CC BY 4.0 license. 

Reviewers' comments:

Reviewer's Responses to Questions

**Comments to the Author**

1. Is the manuscript technically sound, and do the data support the conclusions?

Reviewer #1: Partly

Reviewer #2: Partly

2. Has the statistical analysis been performed appropriately and rigorously?

Reviewer #1: No

Reviewer #2: Yes

3. Have the authors made all data underlying the findings in their manuscript fully available?

Reviewer #1: Yes

Reviewer #2: Yes

4. Is the manuscript presented in an intelligible fashion and written in standard English?

Reviewer #1: No

Reviewer #2: Yes

Reviewer #1: Overall Assessment

This paper addresses a relevant and timely issue — quality of antenatal care (ANC) in Bangladesh — using machine learning (ML) and spatial analysis. The use of BDHS 2022 data ensures national representativeness, and the “hidden hotspot” concept offers practical implications for maternal health policy.

However, the manuscript requires major revision before it can be considered for publication. Key concerns include:

• Lack of clarity and conciseness in writing.

• Inconsistent and sometimes incorrect terminology (e.g., “forecast” vs. “classify”).

• Limited methodological transparency for the ML approaches.

• Logical gaps and unsupported generalizations in the introduction.

• Redundancy and poor sentence structure in results.

• Some interpretation overstates causal inference from predictive models.

General Comments

1. Clarity & Structure

• Many sentences are overly long and repetitive, making it difficult for readers to follow key points.

• Concepts such as “high-quality ANC” are introduced without early definition, requiring the reader to wait until Methods to understand the criteria.

2. Terminology Consistency

• The paper alternates between “forecast” and “predict” without clear distinction. Since the data are cross-sectional, “predict” or “classify” is more accurate.

• The phrase “treatment quality” (line 105) is used interchangeably with “ANC quality,” which is inconsistent.

3. Methodological Transparency

• The justification for using ML over traditional models is weak. The authors should clearly explain why ML is advantageous here (e.g., capturing non-linear interactions, improved predictive performance).

• Data preprocessing steps (handling of missing data, encoding of categorical variables, treatment of class imbalance) are not described.

• Hyperparameter tuning details are minimal — GBM parameters appear to be arbitrarily chosen without validation discussion.

4. Interpretation of Results

• ML feature importance is described as identifying “factors influencing risk,” which could mislead readers into thinking causal effects were estimated. This must be reframed as predictive importance.

• Hotspot analysis thresholds (bottom 40% quartile) are arbitrary; justification or sensitivity analysis is needed.

5. Figures and Tables

• Figures require clearer captions and higher resolution. Captions should be self-contained.

• Tables that span pages should repeat column headers.

6. Policy Relevance

• The discussion should better connect results to actionable policy steps for improving ANC quality in identified hotspots.

Specific Comments by Section

Abstract

• Replace “forecast ANC quality” with “Classify ANC quality” to reflect cross-sectional nature.

• Clearly define “high-quality ANC” briefly in the abstract so readers understand the classification criteria.

• The policy implications could be stated more concisely.

Introduction

1. Lines 55–62:

Global maternal mortality statistics are well presented, but the Bangladesh-specific burden due to inadequate ANC is not quantified. Include national-level evidence linking poor ANC quality to adverse maternal outcomes.

2. Line 64 onwards:

“High-quality ANC” is mentioned without definition until Methods. Define early in the introduction using BDHS/WHO criteria.

3. Lines 68–74:

The paragraph mixes outcomes of high-quality ANC (skilled birth attendance) with components of ANC quality (tests, counseling). Separate for conceptual clarity.

4. Lines 85–88:

The statement about LMICs having increased coverage but disparities in standards is correct but unsupported here. Provide citations, preferably Bangladesh-specific.

5. Lines 105–106:

Replace “treatment quality” with “ANC quality” for consistency.

6. Line 110:

The claim that ML is required to identify women likely to receive inadequate ANC should be softened. Traditional models could also perform this; emphasize ML’s advantages (e.g., non-linearities, interactions).

Methods

1. Lines 172–174:

Replace “generate forecasts” with “predict” or “classify.” Avoid implying temporal forecasting.

2. Statistical vs. ML models:

Which statistical model you used? You only mentions about ML models.

3. Preprocessing:

Describe how categorical variables were handled, how missing data were treated, and whether class imbalance (likely given ~24% high-quality ANC) was addressed.

4. Model tuning:

Provide details on cross-validation, tuning ranges, and rationale for chosen hyperparameters in RF and GBM.

5. Feature importance:

Clarify that ML importance scores indicate predictive contribution, not causal influence.

6. Hotspot analysis:

Justify choice of bottom 40% quartile for “low predicted quality” and discuss potential instability of estimates with small cluster sample sizes.

Results

1. Lines 246–269:

The distribution section is repetitive and overly dense with numbers. Group results logically (age → region → residence → education → wealth → media → facility → decision-making → visits) and use concise sentences.

2. Statistical significance:

When stating differences, indicate whether they are statistically significant and provide p-values or CIs.

3. Figures/Tables:

Ensure all figures have self-contained captions and are readable without referring back to the text.

Discussion

1. Clarify that identified predictors from ML are not causal determinants but predictive markers.

2. Discuss the possibility of unmeasured confounding and the limitations of BDHS cross-sectional design.

3. Expand on how hotspot identification can guide targeted interventions and monitoring.

Reviewer #2: This is a well-structured and relevant research paper that addresses a crucial public health issue in Bangladesh. The use of machine learning models to identify "hidden hotspots" of low-quality antenatal care (ANC) is an innovative approach. The study's findings are significant and provide actionable insights for policymakers. The manuscript clearly outlines the background, methods, results, and conclusions, making it easy to follow.

Strengths

Novelty and Relevance: The study moves beyond simply measuring ANC coverage to evaluating its quality, and introduces the concept of "hidden hotspots," which is highly relevant for targeted interventions. The use of machine learning (ML) models like Gradient Boosting Machine (GBM) is a powerful way to uncover complex relationships and spatial disparities that traditional statistical models might miss.

Clear Methodology: The paper provides a clear description of the data source (BDHS 2022), study population, and the definition of the dependent variable, "Quality ANC". The comparison of three different models (logistic regression, random forest, and GBM) and the use of multiple evaluation metrics (accuracy, sensitivity, specificity, AUC-ROC) demonstrate a robust analytical approach.

Significant Findings: The results highlight that the GBM model performed best and identified key predictors such as the number of ANC visits, wealth index, residence, and education. The finding that women with four or more ANC visits are more than twice as likely to receive high-quality care is a particularly strong and actionable result.

Suggestions for Improvement

1. Enhance the Introduction and Literature Review:

The introduction mentions previous studies that have focused on ANC coverage rather than quality. To strengthen this section, consider providing a more detailed critique of why these traditional methods are insufficient for identifying "hidden hotspots" or the complex interactions of variables.

The paper mentions that "few scholars have looked at the causes and contents of ANC in Bangladesh". While this justifies the study's focus, it would be more impactful to elaborate on what these studies found and how your research either builds upon or diverges from them. For example, Rahman et al. (2017) are mentioned for finding determinants like wealth and education, but the paper could more explicitly state how its ML approach uncovers more nuanced relationships than these earlier studies.

2. Refine the Methodology Section:

The description of the "hidden hotspots" is a key component of the paper, and it could be made more precise. The current definition is "clusters that have a high number of ANC visits... but a reduced expected likelihood of receiving high-quality ANC, specifically the lowest 40% quartile of predicted probability". While this is clear, adding a small-scale example or a hypothetical scenario in the methods section could further clarify this concept for a broader audience.

The paper states that for the Random Forest model, hyperparameters were tuned to achieve the lowest Out-of-Bag (OOB) error rate. While this is a standard practice, including the specific values of

ntree and the optimal mtry that were chosen would add to the reproducibility and transparency of the study.

3. Improve the Results and Discussion:

Table 1: The table is highly detailed and informative. The text in the "Distribution of ANC quality concerning demographic characteristics" section already summarizes the key findings of this table. For better readability, you could consider presenting a concise summary of the most striking results first, followed by a deeper dive into the more nuanced findings.

Statistical Significance: Table 2 provides adjusted odds ratios and p-values from the multivariable logistic regression model. While the text highlights the significant findings, such as the division of Mymensingh, rural residence, and wealth index , it also states that other variables like respondent education and media access "did not exhibit any noteworthy correlations following adjustment". This could be an area for further discussion in the manuscript. It would be valuable to discuss why these variables, which were significant in the descriptive statistics (Table 1), lost their significance in the multivariable model. This could indicate potential confounding effects or multicollinearity, and a brief discussion of this would show a deeper understanding of the data.

4. Strengthen the Conclusion:

The conclusion is succinct but could be expanded to include more specific policy recommendations based on the findings. For example, instead of a general statement that "The findings help policymakers target programs and allocate resources" , the authors could suggest specific programs or resource allocations for the identified "hidden hotspots" in Rangpur and Sylhet. This would transform the conclusion from a summary of the findings into a more impactful call to action.

**Do you want your identity to be public for this peer review?** For information about this choice, including consent withdrawal, please see our Privacy Policy

Reviewer #1: **Yes: ** Md. Ashraful Alam

Reviewer #2: **Yes: ** Ashek Elahi Noor

---

## [Author Response · Author response to Decision Letter 1]

10 Oct 2025

I uploaded a file named "Responses to Reviewers" in which the respected reviewers can find my response.

---

## [Editor Report · Decision Letter 1]

12 Nov 2025

Machine Learning Meets Maternal Health: Uncovering Spatial Blind Spots in Antenatal Care Quality in Bangladesh

PONE-D-25-37518R1

Dear Mr. Chowdhury,

We’re pleased to inform you that your manuscript has been judged scientifically suitable for publication and will be formally accepted for publication once it meets all outstanding technical requirements.

Kind regards,

Russell Kabir, PhD

Academic Editor

PLOS ONE
---

## [Editor Report · Acceptance letter]

PONE-D-25-37518R1

PLOS ONE

Dear Dr. Chakraborty,

I'm pleased to inform you that your manuscript has been deemed suitable for publication in PLOS ONE. Congratulations! Your manuscript is now being handed over to our production team.

Kind regards,

on behalf of

Dr. Russell Kabir

Academic Editor

PLOS ONE